# Antifungal Properties of Biogenic Selenium Nanoparticles Functionalized with Nystatin for the Inhibition of *Candida albicans* Biofilm Formation

**DOI:** 10.3390/molecules28041836

**Published:** 2023-02-15

**Authors:** Shivraj Hariram Nile, Dipalee Thombre, Amruta Shelar, Krithika Gosavi, Jaiprakash Sangshetti, Weiping Zhang, Elwira Sieniawska, Rajendra Patil, Guoyin Kai

**Affiliations:** 1Zhejiang International Science and Technology Cooperation Base for Active Ingredients of Medicinal and Edible Plants and Health, The Third Affiliated Hospital, School of Pharmaceutical Science, Jinhua Academy, Zhejiang Chinese Medical University, Hangzhou 310053, China; 2Department of Biotechnology, Savitribai Phule Pune University, Pune 411007, India; 3Department of Technology, Savitribai Phule Pune University, Pune 411007, India; 4Y. B. Chavan College of Pharmacy, Dr. Rafiq Zakaria Campus, Aurangabad 431001, India; 5Department of Natural Products Chemistry, Medical University of Lublin, Chodzki 1, 20-093 Lublin, Poland

**Keywords:** antibiofilm, nanoparticles, morphogenesis, biogenic selenium, *Candida albicans*

## Abstract

In the present study, biogenic selenium nanoparticles (SeNPs) have been prepared using *Paenibacillus terreus* and functionalized with nystatin (SeNP@PVP_Nystatin nanoconjugates) for inhibiting growth, morphogenesis, and a biofilm in *Candida albicans*. Ultraviolet–visible spectroscopy analysis has shown a characteristic absorption at 289, 303, and 318 nm, and X-ray diffraction analysis has shown characteristic peaks at different 2θ values for SeNPs. Electron microscopy analysis has shown that biogenic SeNPs are spherical in shape with a size in the range of 220–240 nm. Fourier transform infrared spectroscopy has confirmed the functionalization of nystatin on SeNPs (formation of SeNP@PVP_Nystatin nanoconjugates), and the zeta potential has confirmed the negative charge on the nanoconjugates. Biogenic SeNPs are inactive; however, nanoconjugates have shown antifungal activities on *C*. *albicans* (inhibited growth, morphogenesis, and a biofilm). The molecular mechanism for the action of nanoconjugates via a real-time polymerase chain reaction has shown that genes involved in the RAS/cAMP/PKA signaling pathway play an important role in antifungal activity. In cytotoxic studies, nanoconjugates have inhibited only 12% growth of the human embryonic kidney cell line 293 cells, indicating that the nanocomposites are not cytotoxic. Thus, the biogenic SeNPs produced by *P. terreus* can be used as innovative and effective drug carriers to increase the antifungal activity of nystatin.

## 1. Introduction

*Candida albicans* is an opportunistic pathogenic yeast commonly associated with superficial and systemic infections [1]. *C. albicans* has become difficult to kill due to its virulence factors and increasing antifungal resistance [2,3]. The virulence factors include yeast-to-hyphae transition and biofilm formation [2,4,5]. Although yeast and hyphal forms have a role in the pathogenicity of *C. albicans*, the hyphal form of *C. albicans* is the main invasive form [6]. The transition from yeast to hyphae is known as dimorphism [7]. Various factors induce dimorphism, which includes pH, starvation, serum, N-acetylglucosamine, temperature, and CO_2_ [8]. Dimorphism is the first committed step by which *C. albicans* invade the host [5]. The hyphal protrusions formed during dimorphism help *C. albicans* invade the host tissue. Generally, under the appropriate conditions, and after dimorphic, biofilm formation starts; the latter increases the survival rates and traits of multidrug resistance in *C. albicans*, which has become a challenge for therapeutic intervention [9]. A significant proportion of hospital morbidity has been caused by *Candida* invaginating the blood (candidemia) in severe infections. This is especially true with implanted medical devices prone to *C*. *albicans* biofilms. Removing these infected implant devices is often the only option to avoid deadly bloodstream and organ infections. However, this is more expensive and dangerous for the patient. Therefore, there is always a need for new and non-invasive therapeutic strategies. Furthermore, current antifungal drugs are ineffective against *C. albicans* biofilms, because antifungal molecules are absorbed through the extracellular polymers of the biofilm, which have upregulated efflux systems [4]. One of the promising antifungal molecules, nystatin, is used to prevent and treat *C. albicans* for recurrent topical or oral fungal infections [10,11,12]. However, a higher dose of nystatin can cause kidney problems, rashes, nausea, diarrhea, and vomiting. Nystatin also has the drawback of not working against resistant *Candida* strains, thus requiring alternative therapies [12]. The combination of nystatin with nanoparticles (NPs) will likely increase the bioavailability and efficacy of nystatin and also a reduction in its toxicity [13,14]. The NPs have a high surface area, facilitating enhanced drug loading, more efficient drug delivery, and reduced toxicity and dosage for anti-*Candida* activity [15,16,17,18]. The NPs can be used as smart drug delivery systems by encapsulating or attaching the antifungal agents with biocompatible NPs. When using NPs to deliver nystatin, it is important to ensure that the NPs are not cytotoxic. The majority of studies demonstrating the potential of NPs as nystatin carriers have used cytotoxic nanomaterials [19,20,21]. Furthermore, the colloidal stability of the NPs in a solution is another issue for the application of NPs as carriers of nystatin. SeNPs, for example, exhibit good antimicrobial activity and low cytotoxicity toward mammalian cells in addition to their antimicrobial effect. Several chemical methods for synthesizing SeNPs require reducing and stabilizing chemicals that can be toxic when used in biological systems and prevent their efficient utilization [22,23]. In order to improve biocompatibility and stability, SeNPs must be synthesized biologically. Over the last two decades, researchers have become increasingly interested in the green synthesis of NPs [24,25,26,27,28] including SeNPs [23]. Microorganism-mediated methods are regarded as superior to chemical methods when preparing low-cost and nonhazardous biological agents. Biogenic SeNPs are safe to use as drug carriers due to their high degradability, low toxicity, and gradual elimination from the body [29,30].

This study aims to synthesize biogenic SeNPs as biocompatible carriers for nystatin (SeNP@PVP_Nystatin nanoconjugates). Additionally, we have investigated the molecular mechanisms that regulate the antifungal activity of nanoconjugates by real-time polymerase chain reaction (RT-PCR). According to this study’s results, using biogenic SeNPs in conjunction with antimicrobial drugs is beneficial, as the quantities of both agents (biogenic SeNPs and drugs) can be significantly reduced, and the number of adverse side effects can also be reduced. The novelty of this study is the use of biogenic SeNPs as carriers for delivering nystatin, an antifungal drug, to inhibit the virulence factors of *C. albicans*.

## 2. Results and Discussion

### 2.1. Synthesis of Biogenic SeNPs and SeNP@PVP_Nystatin Nanoconjugates

The synthesis of the SeNPs was followed by observing the change in the color of the Tryptic Soy Broth (TSB) broth. We used lab isolates to screen the synthesis of biogenic SeNPs. Among the numerous isolates tested for tolerance toward sodium selenite at various concentrations, *P. terreus* (NCBI accession number KM203682) showed the highest tolerance, up to 5 mM (Appendix A). As seen in Appendix A, the growth kinetics of *P. terreus* during the synthesis of biogenic SeNPs was similar to the control (up to 10 mM sodium selenite) for 24 h; therefore, the *P. terreus* isolate was used to synthesize SeNPs.

After *P. terreus* cells were added to the TSB medium supplemented with 1 mM sodium selenite, the color of the TSB medium changed from yellow to brick red within 6 h, and the intensity increased gradually over 48 h. The red coloration of the TSB was in agreement with previous reports [31,32] and indicated the formation of SeNPs [33,34]. The biogenic SeNPs were orange in color due to the surface plasmon resonance of SeNPs [35]. A UV-Vis spectroscopy observation during biogenic SeNP synthesis is illustrated in Figure 1. The SeNPs exhibited a characteristic peak at around 270 nm [35]. Using UV-Vis spectroscopy, we found that the synthesis of biogenic SeNPs started within 6 h and the synthesis of SeNPs increased with an increase in incubation time. As seen in Appendix A, the colony-forming units of *P*. *terreus* during the synthesis of biogenic SeNPs were nearly the same as the control, indicating that the biogenic SeNPs were produced during the logarithmic growth phase of *P*. *terreus,* which is in agreement with the available reports [36,37,38]. The size of the biogenic SeNPs did not increase with an increase in the incubation period (Appendix A); in other words, there was no linear relationship between the incubation time and the NPs’ size. The relative, unaltered absorption spectra of the formed biogenic SeNPs, between 6 and 48 h, also indicated the stability in the sizes of the produced SeNPs [32]. However, biogenic SeNPs synthesized by using *Pantoea agglomerans* showed a linear relationship between incubation time and NPs size; as the incubation time increased from 15 h to 20 h, the average size of the biogenic SeNPs increased from 90–110 nm to 100–120 nm [39]. As the microbial cells reduced the selenium ions to elemental Se^0^, we excepted a reduction in the concentration of the selenium ions in the broth after separating the synthesized biogenic SeNPs and the cells from the TSB broth (Appendix A). In Appendix A, the concentration of selenium ions decreased linearly for 24 h, indicating that biogenic SeNPs were synthesized linearly during mediated synthesis. The microbial-mediated synthesis of SeNPs was well-documented [40,41,42,43,44,45]. The microbial-mediated synthesis did not require any specialized apparatus or condition. Whole cells or cell extracts of *Bacillus* sp. B2 [43], *Penicillium chrysogenum* [44], *P. expansum* [45], and *Bacillus subtilis* BSN313 [34] were used as reducing agents for the synthesis of SeNPs, with an incubation time of up to five days. However, SeNP biosynthesis using *P*. *terreus* has not been reported yet. Interestingly, unlike five days of incubation time for the synthesis of SeNPs in the aforementioned reports, in the present study, the synthesis of biogenic SeNPs starts within 6 h. Thus, the synthesis of biogenic SeNPs is an efficient and feasible method for the large-scale synthesis of SeNPs. The synthesis of biogenic SeNPs can be summarized as follows [39]
SeO_3_^2−^ + 4e^−^ + 6H^+^ ⇆ Se^0^ + 3H_2_O

*Paenibacillus* bacteria are rod-shaped Gram-positive or Gram-variable, endospore-forming, aerobic, or facultatively anaerobic bacteria. In facultative anaerobes, either the Se oxyanions act as the electron acceptors and form extracellular granules composed of stable, uniform nanospheres (with diameters around 200 nm) of SeNPs having trigonal structures, or the selenite is reduced by a Painter-type reaction in which Se-digluthathione intermediates are formed followed by SeNPs (because selenite is highly reactive with thiol groups) [44]. Following the synthesis of biogenic SeNPs, it is functionalized with nystatin via polyvinylpyrrolidone (PVP) to form SeNP@PVP_Nystatin nanoconjugates. PVP is a biocompatible polymer and is used as a binder to help nystatin to coat the surface of the biogenic SeNPs. Nystatin is a proven antifungal molecule of broad specificity and therefore has been used in the present study. The synthesis of nanoconjugates is shown in Figure 1. As can be seen in Figure 1, the characteristic peak for the biogenic SeNPs shifted from 270 nm to 245 nm in a Nystatin_Composite, due to the presence of polymeric layers, polyvinylpyrrolidone (PVP), which collect the n-electrons of PVP on the surface of biogenic SeNPs [46]. In nanocomposites, three additional peaks (289, 303, and 318 wavelengths) have been observed, which correspond to multiple nystatin functionalization on biogenic SeNPs [47].

### 2.2. Characterization of Biogenic SeNPs and SeNP@PVP_Nystatin Nanoconjugates

Before characterization, the biogenic SeNPs were separated from the culture broth by centrifugation. After centrifugation, the pellets obtained contained cells and NPs, which were first rinsed repeatedly with distilled water, followed by octanol, and then suspended in octanol for 24 h, to separate the NPs from the cells [48]. The nanoconjugates were first characterized by X-ray diffraction (XRD) analysis. Figure 2 shows the XRD analysis of biogenic SeNPs and nanoconjugates. Sharp diffraction peaks at 2θ (degrees) of 23.57°, 29.73°, 41.28°, 43.68°, 45.43°, 51.72°, 56.07°, 61.62°, 65.24°, and 71.60°, respectively, have been indexed as the (100), (101), (110), (102), (111), (201), (112), (103), (210), and (113) planes of Se. All the diffraction peaks can be indexed for the trigonal phase of selenium, which are in good agreement with the reported data (JCPDS card No. 06-362) [49,50]. In nanocomposites, the presence of nystatin over biogenic SeNPs was studied by Fourier Transform Infrared Spectroscopy (FTIR). Figure 3 shows an FTIR analysis of nanoconjugates. The FTIR analysis showed the characteristic peaks for the amide bonds at 1654 cm^−1^ and 1540 cm^−1^, indicating the presence of proteins (from *P*. *terreus*) for the stabilization of the biogenic NPs. A C=O peak at 1665 cm^−1^ can be attributed to the PVP. In FTIR, the nystatin could be ascertained from a broad peak at 3367 cm^−1^, which could be assigned to O-H stretching vibration [51]. The peak located at 2937 cm^−1^ could be attributed to the CH_2_ stretch vibration. The peaks at 1709 and 1620 cm^−1^ could be attributed to C=O stretching vibrations in the carboxylic group and C=C asymmetric stretching, respectively [52]. Furthermore, the peaks at 1069, 848, and 795 cm^−1^ could be attributed to -CH_2,_ and -C-H stretch vibrations [13]. As nystatin was functionalized over the surface of biogenic SeNPs, a significant increase in the band intensity of the C-H bond could be detected at 2922 cm^−1^.

After functionalizing nystatin over the biogenic SeNPs, both biogenic SeNPs and SeNP@PVP_Nystatin nanoconjugates were characterized for size, shape, size distribution, and zeta potentials. The shape and size of biogenic SeNPs and nanoconjugates were determined by Field Emission Scanning Electron Microscopy (FESEM) (Figure 4). As seen in the FESEM micrograph, almost all the NPs were perfect round spheres, which seemed to be a commonly observed feature of biosynthesized NPs, as was evident in other studies [53,54,55]. There was an accumulation of electron-dense particles around the cells or the outer side of the membrane. When the cells were grown in TSB without selenite, these particles did not appear. In spite of the accumulation of Se^0^, with a tendency to accumulate particles into aggregates, the outer membrane did not appear distorted or lysed. The biogenic SeNPs appeared to have a spherical shape and a diameter of approximately 200 nm, similar to those found in *P. motobuensis* [56], *Stenotrophomonas maltophilia* [57], and *Alcaligenes faecalis* [58], *Duganella* sp., *Agrobacterium* sp. [55], *Enterobacter cloacae* [59], *B. selenitireducens* [60], and *Azospirillum brasilense* [61]. After functionalizing nystatin over the biogenic SeNPs, the size of the biogenic SeNPs increased to 240 nm, indicating the successful formation of nanoconjugates.

Dynamic light scattering (DLS) analysis was performed to determine the size range analysis of biogenic SeNPs and nanoconjugates. The elemental analysis of biogenic SeNPs and nanoconjugates (Appendix A) by EDX analysis confirmed the characteristic peak for SeLα at 1.37 keV, along with peaks for carbon, oxygen, and nitrogen [35].

As SeNPs were synthesized by using the whole cell mass and later functionalized with nystatin, we were expecting the charges on biogenic SeNPs and nanoconjugates. The charges were determined by measuring the zeta potential on biogenic SeNPs and nanoconjugates. The zeta potential value for biogenic SeNPs and nanoconjugates were, respectively, −37.44 and −37.14 (Table 1). The negative zeta potential value indicates the colloidal stability of the biogenic SeNPs and nanoconjugates, which is very essential for the biological application of biogenic SeNPs and nanoconjugates [62]. The polydispersity indices further support the colloidal stability of biogenic SeNPs, which is due to the presence of proteins [44,63].

### 2.3. In Vitro Antifungal Activities of Biogenic SeNPs and SeNP@PVP_Nystatin Nanoconjugates

We studied the in vitro antifungal activities in *C. albicans*. Three in vitro antifungal activities of biogenic SeNPs and nanoconjugates were studied: Growth inhibition, morphogenesis inhibition, and antibiofilm. Growth inhibiting activity was performed to know if biogenic SeNPs and nanoconjugates inhibited the growth of *C. albicans* and antibiofilm activity was performed to know if biogenic SeNPs and nanoconjugates inhibit mycelial growth and polysaccharide synthesis. Morphogenesis inhibition was examined to find out whether biogenic SeNPs and nanoconjugates inhibited the transition of yeast to hyphal forms. 

The growth and biofilm inhibition studies were quantitative in nature and performed by measuring the metabolic activities in the yeast and hyphae via the 3-(4,5-Dimethylthiazol-2-yl)-2,5-Diphenyltetrazolium Bromide (MTT) assay, morphogenesis inhibition studies were qualitative in nature and performed by observing the cellular morphology (yeast and hyphae), under (FESEM). First, we identified whether biogenic SeNPs possessed the aforementioned in vitro antifungal activities. Biogenic SeNPs did not inhibit the growth and biofilm of *C. albicans* (Figure 5) till 500 µg/mL. This was very surprising as most of the cited literature for the biogenic SeNPs showed moderate-to-good in vitro antifungal activities. At concentrations between 8 and 512 mg/mL, SeNPs synthesized by *S. maltophilia* and *B. mycoides* inhibited the growth of *Pseudomonas aeruginosa* in clinical isolates, but not *C. albicans* and *C. parapsilosis* [64]. Similarly, another study [65] demonstrated a strong inhibitory effect of SeNPs (10 μg/mL) on the growth of four Gram-positive pathogens, *Staphylococcus aureus*, *B. cereus*, methicillin-resistant *S. aureus*, and *S. agalactiae*. 

Interestingly, other Gram-negative bacteria (*P. aeruginosa*, *Enterobacter* sp., *Enterococcus* sp., *Proteus mirabilis*, *Klebsiella* sp., *Salmonella enteritidis*, and *S. maltophilia*) and *C. albicans* were not inhibited at this concentration. However, higher concentrations of SeNPs (400 μg/mL) that were synthesized by using *B. subtilis* BSN313 showed potential antimicrobial activity on *St. mutans*, *B. cereus*, *Escherichia coli*, and *C. albicans* [36]. A larger MIC concentration of biogenic SeNP (>500 g/mL) on *C. albicans* is comparable to other reports [66]. As biogenic SeNPs have not inhibited the growth and biofilm, we decided to prepare SeNP@PVP_Nystatin nanoconjugates by coating nystatin, an antifungal drug, over the biogenic SeNPs, to use against *C. albicans*. Nystatin is a known antifungal molecule often used to treat *C. albicans* infections. Nystatin is coated over the biogenic SeNPs as a composite (Nystatin_Composites), by using a water-soluble polymer, PVP, made from the monomer N-vinylpyrrolidone. PVP polymers are non-cytotoxic, biodegradable, and do not possess antifungal activity. The purpose of preparing a nanocomposite is to create extended-release carriers either to enhance the delivery or uptake of nystatin into the target cells or to reduce the toxicity of the free nystatin.

The growth and biofilm inhibition of the biogenic nystatin, SeNPs, nystatin-PVP, and nanocomposite are shown in Figure 5. Nystatin has shown antifungal activity, and the activity increases with increasing concentrations of nystatin. Similarly, the Nystatin_PVP composite has also shown antifungal activity comparable to nystatin. However, nanocomposites have shown improved antifungal activity against *C. albicans*. The calculated MIC_50_ value for the growth inhibition for nystatin and PVP_Nystatin was between 7.8 to 15.6 µg/mL. Similarly, the calculated MIC_50_ value for the antibiofilm activity of nystatin was 31.2 µg/mL. The MIC_50_ value for the growth inhibition and antibiofilm activity for nanoconjugates had decreased to less than 3.9 µg/mL. A possible explanation for these increased in vitro antifungal activities of nanoconjugates may be because of the greater surface area of NPs, which enhances nystatin’s solubility and availability [67]. Another reason could be the localization of nanocomposites over the surfaces of cells. By directly contacting the surface of cells, nanocomposites may effectively recruit the nystatin molecules to enhance the antifungal activity. 

As a result of the intercalation of nystatin into ergosterol-containing membranes of the fungal cells, it forms channels that prevent ion transfer in the cell, causing the leakage of cytoplasmic contents and eventually cell death. The biodegradable nature of PVP further assists the antifungal activity by the slow release of nystatin molecules on or near the surface of cells, which is well-documented in the literature [68,69]. The nystatin that functionalized over the surface magnetic NPs was shown to possess more fungicidal activity than unbound nystatin, due to the enhanced ability of the magnetic NPs to improve nystatin penetration, thus improving their killing properties and exerting a rapid effect on the *Candida* cells [70]. The nystatin coated on iron NPs and the chitosan composite [71], as also the nystatin-conjugated iron oxide nanocomposite [72] have also shown improved antifungal activity over unbound nystatin. The antifungal activity of nanocomposites has become more evident following studies on morphogenesis inhibition. At 3.9 µg/mL, nanoconjugates inhibited the morphogenesis of the yeast form to the hyphae form (Figure 6B,D). Control samples (without nanocomposites) showed the transition of the yeast form to the hyphal form.

Finally, the mechanism of the antifungal activity of nanoconjugates was studied by measuring the transcript levels of important genes involved in morphogenesis and the biofilm formation in *C. albicans*, using RT-PCR. Essentially, the RAS/cAMP/PKA signaling pathway was studied, as the RAS/cAMP/PKA pathway played an important role in yeast to hyphae transition and biofilm formation. Adhesion was the first step for the initiation of biofilm formation in *C. albicans* and was mediated by the expression of *Als* genes, which encoded the adhesion proteins that helped the cells to adhere to the hydrophobic surfaces or to attach themselves to the endothelial and epithelial cells [73,74,75]. *Als*3, 4, and 6, and the *Hwp*1 genes were downregulated in the presence of nanoconjugates (Figure 7). The *Hwp*1 gene that was involved in hyphal growth and biofilm formation [76] was also downregulated. The *Nrg*1 and *Tup*1 genes acted as *Hwp*1 repressors [77], and both *Nrg*1 and *Tup*1 were upregulated. The *Efg*1 gene played an important role in hyphal growth and biofilm formation, and its expression was related to the *Als* and *Hwp*1 gene expressions [78]. The downregulation of the *Efg*1 gene likely downregulated the expression of the hyphal cell-wall genes, such as *Hwp*1 and *Als*3. The *Phr*1 gene expression was seen to inhibit yeast-to-hyphae transition, and the downregulation of the *Phr*1 gene likely inhibited the morphogenesis in *C. albicans* in the presence of nanoconjugates.

Finally, to confirm the potential application of biogenic SeNPs, cytotoxic studies were performed on HEK-293 cells. The biogenic SeNPs were not cytotoxic on HEK-293 until 125 µg/mL (Appendix A), thus paving the way for its use as a carrier for the effective delivery of the antifungal drug, nystatin, against *C. albicans*. Thus, the present study utilizes the application of biogenic SeNPs as a vehicle to deliver the effective concentration of nystatin for antifungal activities. The present study is significant as it decreases the effective concentration of the nystatin required for antifungal activities. In the future, biogenic SeNPs can be used as carriers to deliver other antimicrobial molecules.

## 3. Materials and Methods

The media components used for the experiments were purchased from Hi-Media (Mumbai, India). The chemicals, such as sodium selenite, nystatin, PVP, dimethyl sulfoxide (DMSO), and other reagents used in the experiments were of analytical grade and purchased from Sigma-Aldrich (Bangalore, India). *C. albicans* ATCC 227 was a kind gift from Dr. Zore, SRTMU, Nanded.

### 3.1. Screening of Selenium-Tolerant Bacteria

The isolates selected for the screening of selenium-tolerant bacteria were the lab isolates collected from the Western Ghats of Maharashtra, India. To put it briefly, the isolates with 1.0 optical density (O.D) were inoculated into the tryptic soy broth (TSB) containing various concentrations (1 to 75 mM of sodium selenite) at 37 degrees centigrade (°C) and 150 rotation per minute (rpm) with 24 h of incubation. Following the incubation, an aliquot of the samples was spread-plated, and the plates were incubated at 37 °C for 24 h for the colonies to appear. Any spread-plated plate (containing sodium selenite) that did not show any colonies was regarded as having minimum inhibitory concentrations (MIC) of selenium against the tested isolates. The isolate that tolerated the maximum concentrations of sodium selenite was identified by using 16S rRNA sequencing and later used for the synthesis of SeNPs (biogenic SeNPs).

### 3.2. Synthesis of Biogenic SeNPs

*P. terreus* was an isolate that was found to be resistant to sodium selenite, and this isolate was used to biosynthesize the SeNPs. In short, 100 mL of the TSB nutrient medium supplemented with 1 mM of sodium selenite in a 250 mL Erlenmeyer flask was inoculated with 1% *v/v* of *P. terreus* and grown overnight. The TSB media with 1 mM sodium selenite was used as the control. Flasks were incubated at 37 °C, at 150 rpm. Samples were aseptically withdrawn at different time intervals. SeNPs were separated from cells, purified, and stored in water at 4 °C for further analysis [12].

### 3.3. Synthesis of SeNP@PVP_Nystatin Nanoconjugates

The synthesis of nanoconjugates was performed by functionalizing nystatin on biogenic SeNPs. An aqueous suspension of 10 mg/mL of the SeNPs solution was prepared and sonicated at 40 Hz for 30 min. Thirty milliliters of 0.5 mg/mL PVP solution was added dropwise into the biogenic SeNP suspension, with stirring conditions at 50 °C for 1.5 h, followed by the dropwise addition of 15 mL of 1 mg/mL of nystatin solution. This total mixture was heated at 50 °C until all the alcohol evaporated (2–3 h), and then it was cooled and centrifuged at 10,000 rpm for 10 min and finally air-dried so the nanoconjugates could be used for further study.

### 3.4. Characterization of Biogenic SeNPs and SeNP@PVP_Nystatin Nanoconjugates

Crystallographic characterization of biogenic SeNPs and nanoconjugates was performed by XRD analysis. The XRD measurements were performed with the help of the Bruker D8 Advance Diffractometer (Bruker, Germany) (40 kV, 40 mA, Cu-Kα, λ = 1.54 Å) using 2-theta values ranging from 10 to 90. Biogenic SeNPs and nanoconjugates were characterized by UV-Vis spectroscopy (Shimadzu 1800, Kyoto, Japan) and FESEM (FEI Nova NanoSEM 450) for determining the size and shape of the NPs. Elemental composition was performed by using an EDX (Bruker XFlash 6I30) micro-analysis system coupled with the FESEM.

### 3.5. Antifungal Activity Assay of Biogenic SeNPs and SeNP@PVP_Nystatin Nanoconjugates

The standard broth micro-dilution method was used to assay the antifungal activity (growth inhibition) of biogenic SeNPs and nanoconjugates on the *C. albicans* in the planktonic form. In short, the concentration of each of the biogenic SeNPs and nanoconjugates in a range of 3.9 to 500 µg/mL was added to 100 µL of the Roswell Park Memorial Institute (RPMI)-1640 medium, containing 1 × 10^4^ cells/mL, in 96-well plates (Costar, Corning Inc., Corning, NY, USA). The plates were incubated at 37 °C for 48 h, and the growth was measured by taking absorbance at 620 nm using a microplate reader (Multiskan EX, Thermo Electron Corp., Waltham, MA, USA). The lowest concentration of each biogenic SeNP and nanoconjugate, which caused a 50% reduction in the absorbance compared to the control, was considered as MIC_50_. All the experiments were performed in triplicate.

### 3.6. Anti-Biofilm Assay of Biogenic SeNPs and SeNP@PVP_Nystatin Nanoconjugates

An anti-biofilm assay was performed in tissue culture-treated, 96-well, polystyrene plates. In short, 100 μL of 1 × 10^7^ cells /mL suspensions were added to 96-well plates and allowed to adhere to a solid surface at 37 °C for 90 min. After adhesion, the unbound cells were removed by washing twice with phosphate-buffered saline (0.1 M, pH 7.0). The wells were reconstituted with RPMI-1640 medium (200 μL) along with various concentrations of biogenic SeNPs and nanoconjugates separately, and the plates were incubated for 48 h at 37 °C. The developed biofilms were observed under an inverted light microscope (Metzer, India) and quantified by a 3-(4,5-Dimethylthiazol-2-yl)-2,5-Diphenyltetrazolium Bromide (MTT) assay. All the experiments were performed in triplicate.

### 3.7. Anti-Morphogenesis Assay of Biogenic SeNPs and SeNP@PVP_Nystatin Nanoconjugates

An anti-morphogenesis assay of biogenic SeNPs and nanoconjugates was performed in 20% serum. In short, various concentrations of each biogenic SeNP and nanoconjugate were added into 100 μL of 20% serum-containing 1 × 10^6^ cells/mL of *C. albicans* and incubated for 90 min at 37 °C in an orbital shaker at 200 rpm. After incubation, the cells were observed for the formation of the germ tube (hyphae) under a microscope (Labomed microphotography system (Labomed, India)) at 200 × magnification. The inhibition of 50% hyphae formation was compared with the control and considered as the MIC_50_ for morphogenesis. All the experiments were performed in triplicate. For observing the morphologies of cells in the presence of MIC_50_ values of biogenic SeNPs and nanoconjugates, the cells were fixed in 2.5% of glutaraldehyde in 0.1 M PBS (pH 7.2) for 24 h at 4 °C and then dehydrated in a series of graded alcohols. The cells were preserved in 100 μL absolute alcohol. Finally, the cells were layered onto a glass, coated with gold ions, and viewed under FESEM (FEI Nova NanoSEM 450).

### 3.8. Anti-Morphogenesis Mechanism of SeNP@PVP_Nystatin Nanoconjugates

The anti-morphogenesis mechanism action of nanoconjugates was studied by measuring the differential expression of the hyphae by using RT-PCR. In short, the *C. albicans* cells (1 × 10^7^ cells/mL) were inoculated in a 20% serum-containing MIC concentration of SeNPs and nanoconjugates and incubated for 90 min at 37 °C. The total RNA was isolated by using the RNeasy^®^ Mini Kit (QIAGEN) and converted to cDNA by using Superscript^®^ III for the first-strand synthesis (Invitrogen, Life Technologies, Carlsbad, CA, USA). The PCR reactions were conducted by using the SYBR Green Real-Time PCR Master Mixes, Thermofisher scientific, Waltham, MA, USA) with preliminary denaturation for 3 min at 95 °C. This was followed by 32 amplification cycles of denaturation at 95 °C for 30 s and annealing at 60 °C for 20 s with primer extension at 72° C for 30 s (CFX 96 Real-time System, Bio-Rad, Hercules, CA, USA). Actin was used as an internal control, and the transcript levels of the selected genes were calculated using the formula 2^−ΔΔCT^ [79]. All the reactions were run in triplicate using biological replicates, and the experiment was repeated thrice. Data were reported as mean ± standard deviation (S.D.). The gene expression was normalized to ACTIN gene levels.

### 3.9. Cytotoxicity Study of Biogenic SeNPs and SeNP@PVP_Nystatin Nanoconjugates on HEK-293

The cytotoxicity effects of biogenic SeNPs and nanoconjugates were studied by using the HEK-293 cell line. In short, 100 μL of Dulbecco Modified Eagle Medium (DMEM) was added into 96-well plates with 1 × 10^3^ cells per well. Biogenic SeNPs and nanoconjugates, in the range of 3.9 to 125 µg/mL each, were added separately into the DMEM and incubated at 37 °C with 5% CO_2_ for 24 h. After incubation, the quantitative analysis of the effect of biogenic SeNPs and nanoconjugates was performed with the help of an MTT assay, as mentioned earlier.

## Figures and Tables

**Figure 1 molecules-28-01836-f001:**
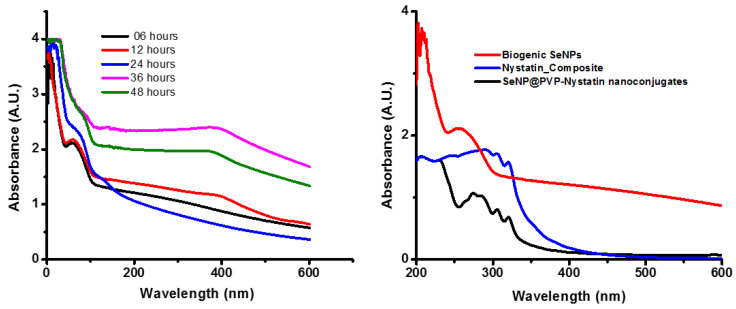
Ultraviolet–visible (UV-Vis) spectra of biogenic SeNPs, Nystatin_Composite, and SeNP@PVP_Nystatin nanocomposites. The absorption peak at 270 nm for the synthesis of biogenic SeNPs at different time interval periods shows that the synthesis of biogenic SeNPs increases with an increase in the incubation period (left); absorption peaks of biogenic SeNPs upon functionalization with nystatin (SeNP@PVP_Nystatin nanoconjugates) show additional peaks at 289, 303, and 318, which are characteristic peaks for nystatin. The Nystatin_Composite also shows nystatin characteristic peaks.

**Figure 2 molecules-28-01836-f002:**
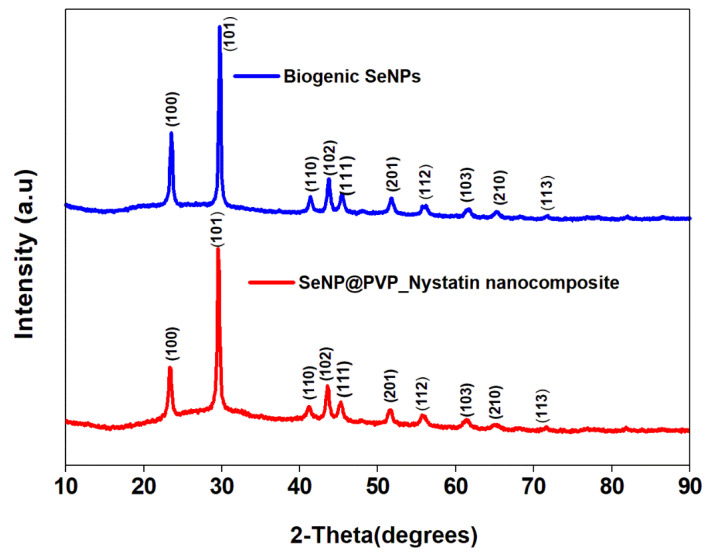
XRD analysis of biogenic SeNPs and SeNP@PVP_Nystatin nanocomposites, showing characteristic 2θ values indexed at the (100), (101), (110), (102), (111), (201), (112), (103), (210), and (113) planes.

**Figure 3 molecules-28-01836-f003:**
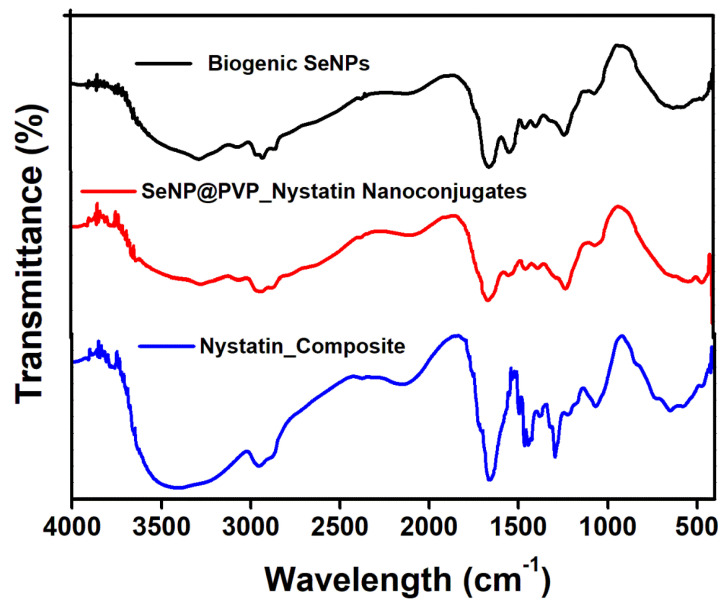
FTIR analysis of Biogenic SeNPs, Nystatin composite, and SeNP@PVP_Nystatin nanoconjugates.

**Figure 4 molecules-28-01836-f004:**
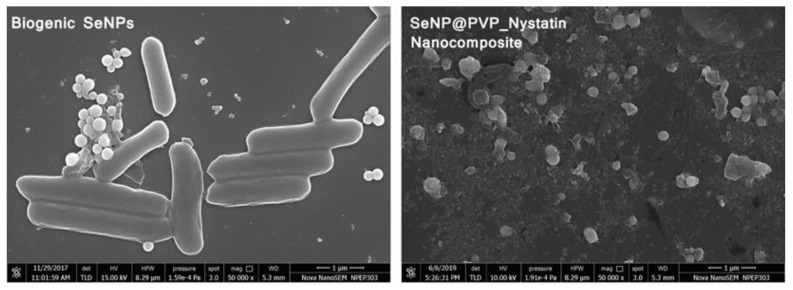
FESEM analysis of Biogenic SeNPs and SeNP@PVP_Nystatin nanoconjugates. Biogenic SeNPs are shown along with cells.

**Figure 5 molecules-28-01836-f005:**
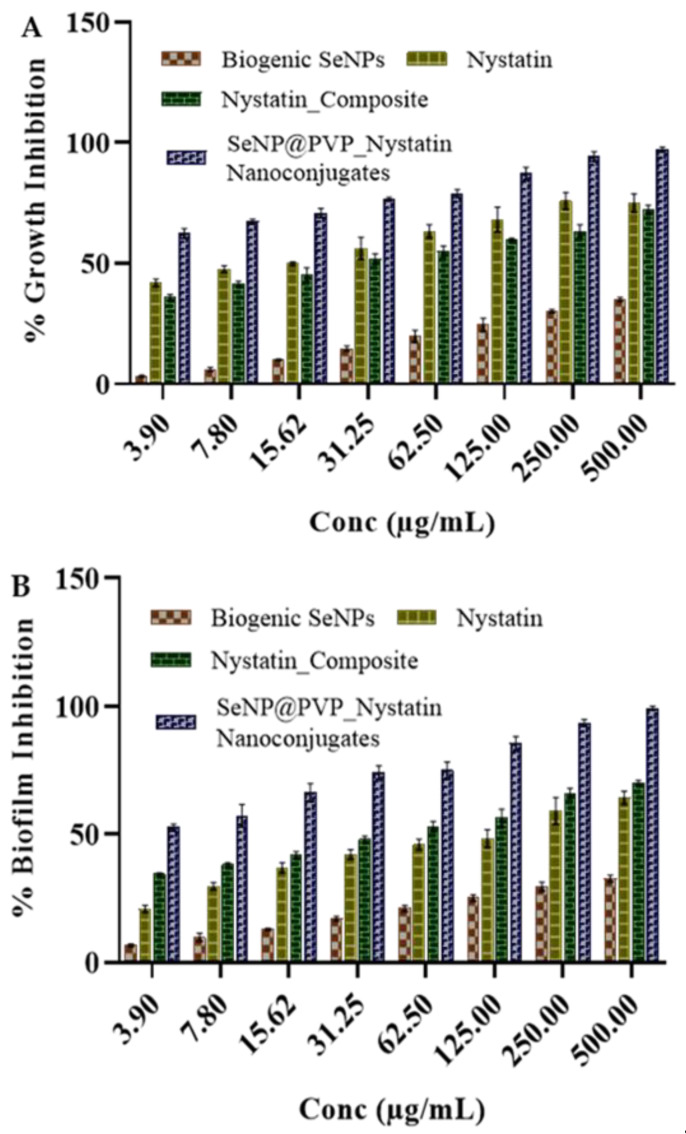
Antifungal activity of biogenic SeNPs, Nystatin_Composites, and SeNP@PVP_Nystatin nanoconjugates on *C. albicans*. Nanoconjugates showed better antifungal activity (inhibition of growth (**A**) and biofilm (**B**)) than composites, and biogenic SeNPs did not show growth inhibition activity up to 500 μg/mL.

**Figure 6 molecules-28-01836-f006:**
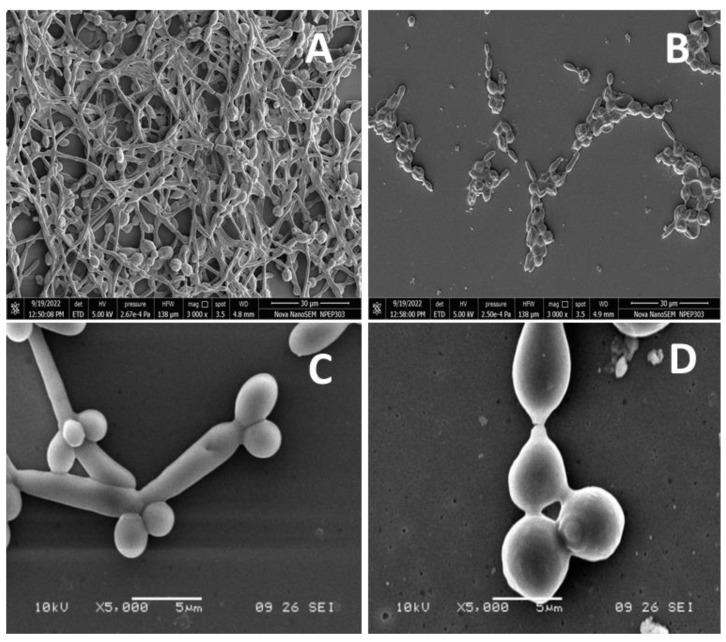
The anti-morphogenesis activity of SeNP@PVP_Nystatin nanoconjugates. Nanoconjugates inhibited yeast to hyphae transitions (**B**,**D**). The control samples (**A**,**C**) showed yeast-to-hyphae transitions.

**Figure 7 molecules-28-01836-f007:**
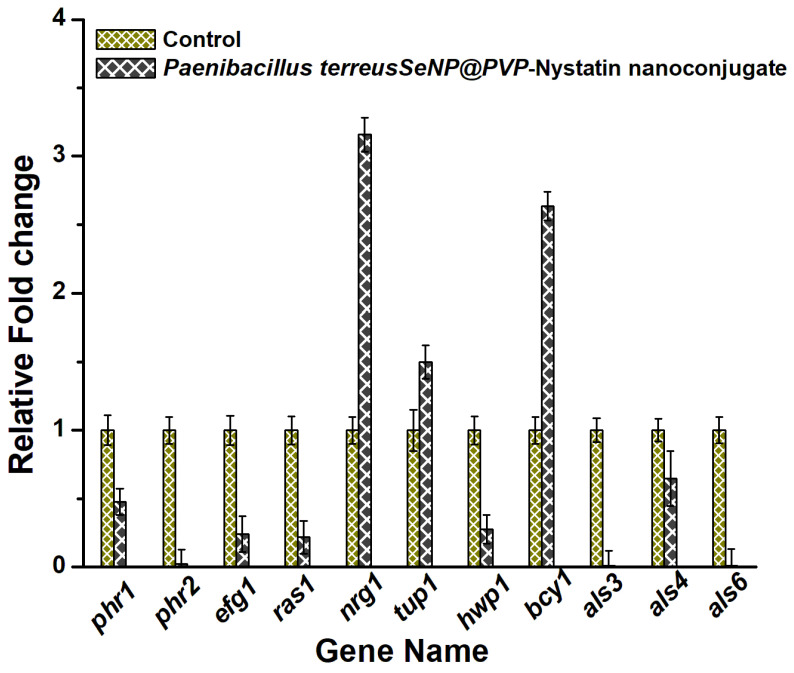
The anti-morphogenesis mechanism of SeNP@PVP_Nystatin nanoconjugates. The real-time polymerase chain reaction (RT-PCR) of the genes (phr1, phr2, efg11, ras11, tup1, hwp1, als3, als4, and als6) involved in the morphogenesis was downregulated in the presence of the nanocomposites.

**Table 1 molecules-28-01836-t001:** DLS, Zeta Potential analysis.

Nanoparticle	DLS	Polydispersity Index	Zeta Potential
Biogenic SeNPs	200–220 nm	0.265	−37.44
SeNP@PVP_Nystatin nanoconjugates	220–242 nm	0.289	−37.14

## Data Availability

Data available on request from authors.

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
