# Peer review of "Antifungal Properties of Biogenic Selenium Nanoparticles Functionalized with Nystatin for the Inhibition of Candida albicans Biofilm Formation"

_molecules, 2023, doi:10.3390/molecules28041836_

Round 1
Reviewer 1 Report
The authors evaluated the antifungal activities of biogenic SeNPs and SeNP@PVP-Nystatin nanoconjugates. The results are interesting. However, some explanations should be revised:
1- The manuscript contains several typographical as well as grammatical errors. Authors are suggested to get the manuscript proofread by a native English speaker.
2- Abstract should be re-written and major experimental results should be provided.
3- The introduction can be improved by providing a more critical discussion of recent related literature. Discuss the shortcomings of previous work and the gaps, and how this work intends to fill those gaps. For example, some papers (Applied Physics A (127(11) (2021) 1-8, 128 (3) (2022) 1-13, and 128 (8) (2022) 1-10), https://doi.org/10.1007/s13204-022-02705-1, and https://doi.org/10.1016/j.ceramint.2022.11.329) should be cited. The authors should determine the novelty of the current research in the last paragraph of the introduction part.
4- # Results and discussion: “Over the last two decades, researchers have become increasingly interested in the green synthesis of metal NPs ….”, it should be placed in the introduction section. Furthermore, the discussion part must be significantly improved.5- The FESEM result of biogenic SeNPs have not been evaluated. What are the effects of PVP-Nystatin on changing the shape of the SeNPs? Explanation needed.
6- The author should propose the interactive relation between the bio-components and SeNPs that give the studied properties and applications. 7- The author should confirm the exact bio-components type and quantity that still attached on SeNPs surface, because such compounds could not be able to detect by FESEM.
8- In FTIR, more bands are not identified or discussed in the text. The authors have not supported their findings by any research studies.
9- What is the role of PVP-Nystatin in the antifungal activities of the SeNPs? Explanation needed.
Author Response
Dear Reviewer,
Thanks for giving us opportunity to revise the manuscript. The authors are grateful to the editors and reviewers for their valuable suggestions and comments that helped to improve the manuscript further. As per the suggestions of the reviewers, we have made necessary changes wherever applicable and such major changes are highlighted in yellow colors. Please find attached copy as Responses to Reviewer Comments # 1

Reviewer 2 Report
1. The overall theme, data and scientific presentation of paper is very good and can be publish after a revision.
2. There are numerous syntax errors across the text. As an example errors are highlighted in “Abstract” in red. Need to recheck syntax errors carefully.
The present study investigated the effectiveness of biogenic selenium nanoparticles (SeNPs) prepared by using Paenibacillus terreus and functionalized with nystatin (SeNP@PVP_Nys-26 tatin nanoconjugates) to inhibit the growth, morphogenesis, and biofilm of Candida albicans. The nanoconjugates were characterized by X-ray diffraction analysis. Field emission scanning electron microscopy and transmission electron microscopy were used for its morphology and size; fourier transform infrared spectroscopy were used for confirming its functionalization with nystatin; and zeta potential was determined for the charge on SeNPs. Biogenic SeNPs and nanoconjugates antibiofilm mechanisms were investigated using real-time polymerase chain reaction (RT-PCR) by measuring the fold expression of gene products crucial to C. albicans biofilm formation. The cytotoxic studies of the nanoconjugates were studied by using human embryonic kidney cell line 293. The nanoconjugates inhibited 80 percent biofilm of C. albicans. The electron microscopic scan of biofilm treated with nanoconjugates provided additional evidence of antibiofilm activity of nanoconjugates. The RT-PCR results showed that genes associated with attachment, morphogenesis, and biofilm formation in C. albicans were severely affected in presence of nanoconjugates. Additionally, the nanoconjugates exhibited a non-cytotoxic effect on cell lines. Therefore, the biogenic SeNPs produced by P. terreus can be used as a smart and effective drug carrier to increase the antifungal activity of nystatin.
3. Do you think “Paenibacillus terreus” is safe to produce SeNPs for their further therapeutic and nutritional application?
4. In section “3.2. Synthesis of biogenic SeNPs” the purification of SeNPs from the Paenibacillus terreus cells is not clear. Give details
5. “In line # 310 the sentence would be corrected as “In short, to a sonicated 50 mL of 10 mg/mL SeNPs suspension, dropwise 30 mL …..” . Secondly, in which solvent the suspension of SeNPs was made? How you adjusted the concentration of SeNPs 10 mg/mL? Not clear.
6. How you did confirm that all Nystatin was conjugated with SeNPs?
Author Response
Dear Reviewer,
Thanks for giving us opportunity to revise the manuscript. The authors are grateful to the editors and reviewers for their valuable suggestions and comments that helped to improve the manuscript further. As per the suggestions of the reviewers, we have made necessary changes wherever applicable and such major changes are highlighted in yellow colors. Please find attached copy as Responses to Reviewer Comments # 2

Round 2
Reviewer 1 Report
This paper has been revised and could be accepted.
Author Response
Dear Reviewer
Thank for your time and suggestions on our manuscript, we carefully gone through our manuscript and revised every point and section as suggested.
Also, we edited manuscript through proper professional native English editor and attaching herewith certificate from editor.
Regards
Prof. Kai